# Complementary and Integrative Medicines as Prophylactic Agents for Pediatric Migraine: A Narrative Literature Review

**DOI:** 10.3390/jcm10010138

**Published:** 2021-01-03

**Authors:** Gaku Yamanaka, Kanako Kanou, Tomoko Takamatsu, Mika Takeshita, Shinichiro Morichi, Shinji Suzuki, Yu Ishida, Yusuke Watanabe, Soken Go, Shingo Oana, Hisashi Kawashima

**Affiliations:** Department of Pediatrics and Adolescent Medicine, Tokyo Medical University, Tokyo 160-0023, Japan; kanako.hayashi.0110@gmail.com (K.K.); t-mori@tokyo-med.ac.jp (T.T.); jerryfish_mika@yahoo.co.jp (M.T.); s.morichi@gmail.com (S.M.); shin.szk@gmail.com (S.S.); ishiyu@tokyo-med.ac.jp (Y.I.); vandersar_0301@yahoo.co.jp (Y.W.); soupei59@gmail.com (S.G.); oanas@tokyo-med.ac.jp (S.O.); hisashi@tokyo-med.ac.jp (H.K.)

**Keywords:** migraine, riboflavin, coenzyme Q10, magnesium, melatonin, polyunsaturated fatty acids, feverfew, vitamin D, ginkgolide B

## Abstract

Complementary and integrative medicines (CIMs) are increasingly used as a preventive antimigraine therapy. In this review, we aimed to summarize the evidence for the efficacy and safety of eight CIMs (riboflavin, coenzyme Q10, magnesium, melatonin, polyunsaturated fatty acids, and combination therapy of feverfew, vitamin D, and ginkgolide B) in pediatric migraine prevention. The level of evidence for riboflavin was relatively high; it was investigated by many studies with five/seven studies demonstrating its efficacy. Five studies investigated the use of melatonin, with one reporting negative results. There was insufficient evidence on the effectiveness of coenzyme Q10, magnesium, and polyunsaturated fatty acids. Combination therapy showed positive potential; however, reports on the individual antimigraine effects of the CIMs were lacking. A definitive conclusion was not reached regarding the specific integrative drugs clinicians should choose for pediatric migraines, owing to low-quality evidence and a limited number of studies. Integrative medications are becoming more common for pediatric migraine prevention as they do not produce serious side effects, and underlying research data suggest their efficacy in preventing migraine. Additional studies are warranted to confirm the role of CIMs in treating patients with migraines.

## 1. Introduction

Migraine headaches are common, occurring in 3–5% of young children and up to 18% of adolescents [1]. Migraine incidence seems to peak earlier in men than in women [2,3], and the sex ratio varies considerably with age; although this ratio is approximately equal before puberty [3,4], across the lifespan, women are two to three times more likely to experience migraine than men [4]. It adversely affects the quality of life like childhood cancer, heart disease, and rheumatic disease [5]. However, to date, there has been no consensus regarding migraine prophylaxis in children and adolescents, even in recently published practice guidelines from the United States [6].

In recent years, complementary and integrative medicines (CIMs) have been increasingly used for health maintenance and treating diseases, such as migraine in children and adolescents [7]. A study involving pediatric headache patients who were referred to an Italian tertiary care headache clinic revealed that 76% of the patients were using CIM therapies for their headaches, with 74% using herbal remedies and 40% using vitamin and mineral supplements [8].

In 2017, the largest and most methodologically rigorous trial on the efficacy of pharmacological interventions for the prevention of migraine in children and adolescents (the Childhood and Adolescent Migraine Prevention Trial (CHAMP)) was published [9]. In this study, both amitriptyline and topiramate, which are widely used as preventive treatments for migraines in children, were determined to be ineffective. Moreover, the study could not be completed because of adverse effects [9]. Based on the results of the CHAMP trial, many clinicians now use nutritional supplements, which cause minimal side effects, as their first-line treatment prior to the use of pharmaceuticals [10]. Therefore, the use of CIMs is expected to increase in the future. The following integrative medicines have been previously used for preventing migraine: riboflavin, coenzyme Q10, magnesium, melatonin, polyunsaturated fatty acids (PUFAs), feverfew, butterbur, vitamin D, and ginkgolide B. In most cases, they have been associated with fewer side effects than pharmaceuticals; however, while butterbur has been reported to be effective in preventing migraine in the pediatric population [11,12], post-marketing data have demonstrated an association with hepatotoxicity [13]. Therefore, classifying a medication as a CAM does not necessarily imply that it is safe; individual efficacies and mechanisms of action may vary widely.

Although CIMs are frequently used for the prophylactic treatment of pediatric headaches, there is a lack of official guidelines and an established list of CIMs that are effective in preventing childhood migraines. Therefore, in this review, we aimed to summarize the evidence for the efficacy and safety of CIMs in the prevention of pediatric migraines.

## 2. Materials and Methods

This narrative literature review was based on studies of the effectiveness of dietary supplements for the prophylactic treatment of migraine in children and adolescents. The literature search was conducted using the PubMed database and included articles published up to December 2020. Observational studies, randomized controlled trials (RCTs), systematic reviews, and meta-analyses were included, while individual case reports were excluded. Only studies involving participants aged under 18 years were included. The following CIMs were analyzed in this review: riboflavin, coenzyme Q10, magnesium, melatonin, PUFAs, and a combination of feverfew, vitamin D, and ginkgolide B. 

## 3. Results

### 3.1. Riboflavin

Riboflavin plays a crucial role in the production of adenosine triphosphate, as it is a precursor for two coenzymes that are required for oxidation-reduction reactions during mitochondrial energy production [14]. Previous studies have described the role of mitochondrial dysfunction in migraine pathophysiology [15,16]. Magnetic resonance spectroscopy studies have revealed reduced interictal rates of mitochondrial oxidative phosphorylation in the brains of patients with migraine [16,17,18]. Recently, the neuroprotective potential of riboflavin to ameliorate oxidative stress, neuroinflammation, and glutamine excitotoxicity through various mechanisms has been reported [19]. 

Two previous systematic reviews that evaluated the role of riboflavin in the prevention of migraine assessed the same studies (three RCTs and one retrospective study) [20,21]. Among these studies, two—one RCT and one retrospective study—reported that riboflavin was effective [22,23], while the other two reported ineffectiveness [24,25]. The latter two studies concluded that there was a lack of evidence to support the use of riboflavin for pediatric migraine. However, the former two positive studies have recently been published [23,26]. 

The included studies are summarized in Table 1.

#### 3.1.1. Negative Studies

The lack of effectiveness of riboflavin for pediatric migraine prevention was reported by two RCTs [24,25]. In the first study that was published in 2008, 48 children and adolescents with episodic migraines were prescribed 200 mg of riboflavin or a placebo for 12 weeks. The migraine headache frequency was not reduced with riboflavin (4.4 headaches/month with riboflavin vs. 4.2 headaches/month with the placebo). The primary outcome was the proportion of patients who exhibited a 50% or greater reduction in headache frequency; 14 of the 21 patients in the placebo group (66.6%) vs. 12 of the 27 patients in the riboflavin group (44.4%) exhibited a 50% reduction (*p* = 0.125) [24]. 

The RCT conducted by Bruijn et al., which included 42 participants with migraines (6–13 years old), also reported a lack of effectiveness. Each child underwent 16 weeks of riboflavin therapy and 16 weeks of placebo therapy, separated by a four-week washout period. The primary outcome of the study was the reduction in the frequency of migraine and tension-type headaches as compared to baseline. The reduction of the mean headache frequency with riboflavin use (3.60 headaches/month to 2.05 headaches/month) was not statistically different from that with placebo use (3.05 headaches/month to 1.4 headaches/month) (*p* = 0.44) [25]. Bruijn et al. selected a riboflavin dosage of 50 mg for their study because an adult migraine study indicated equivalent effects between low (25 mg)- and high (200 or 400 mg)-doses of riboflavin supplementation [46]. The results of this trial may have been influenced by the low dose of riboflavin. Further, the small number of participants may have contributed to the negative results reported in both RCTs. 

#### 3.1.2. Positive Studies

In a retrospective study conducted by Condo et al. in 2009 [27], the efficacy of 200 or 400 mg of riboflavin supplementation for three to six months was evaluated in 41 children and adolescents. The participants in that study were suffering from a variety of headache types: 39 exhibited typical migraine headaches (seven of whom had tension-type headaches) and two had basilar-type migraines and a combination of benign paroxysmal childhood vertigo and frequent episodic tension-type headaches. A statistically significant reduction in headache frequency was observed over the first three months of treatment; however, the efficacy of riboflavin did not persist among the subgroup of patients who underwent four to six months of treatment [27]. 

The effectiveness of 200 mg of riboflavin for the prevention of pediatric migraine was demonstrated in a three-month RCT that included 98 adolescent patients (aged 12–19 years old) [22]. The frequency of headaches decreased from the first month (6.4 episodes/month) to the second (3.9 episodes/month) and third (3.7 episodes/month) months in the riboflavin group. Headache duration in the riboflavin group was lower than that in the placebo group in the second and third months (*p*=0.012 and *p* = 0.001, respectively). In addition, scores on the Pediatric Migraine Disability Assessment Scale (PedMIDAS), a measure of overall functional disability, were significantly decreased in the riboflavin group (*p* = 0.001). 

A recent prospective study reported that riboflavin was effective in reducing the frequency of pediatric migraine by at least 50% over 12 weeks. Among the patients who were prescribed 100 mg/day of riboflavin (*n* = 30), 400 mg/day of riboflavin (*n* = 30), or a placebo (*n* = 30), only the participants who were prescribed 400 mg of riboflavin exhibited a significantly decreased headache frequency (9.3 to 2.9 episodes/month) and duration (7.9 to 3.9 h/month) at 12 weeks compared to baseline. There were no significant differences between the low-dose and placebo groups in terms of frequency (*p* = 0.49) or mean duration (*p* = 0.69) of the migraine attacks [23].

Low-dose riboflavin supplementation (10 or 40 mg) was observed to be effective in our recent retrospective study [26]. Our data revealed a significant overall reduction in the median frequency of headache episodes from baseline to three months (median (interquartile range), 5.2 (3–7) vs. 4.0 (2–5), respectively, *p* = 0.0001). Twenty-five patients (36.7%) exhibited a 50% or greater reduction in episode frequency (responders), while 18 (26.5%) exhibited a 25–50% reduction (semi-responders). The patients also experienced frequent episodic or chronic tension-type headaches, medication overuse headaches, retinal migraines, migraines with brainstem aura, and hemiplegic migraines. Non-responders (63%), including semi-responders, were more likely to have comorbid non-migraine headaches (odds ratio, 4.11; 95% confidence interval (CI), 1.27–13.33; *p* = 0.02); this variable was also significant in a multivariate analysis (adjusted odds ratio, 3.8; 95% CI, 1.16–12.6; *p* = 0.03). Our recent multivariate analysis demonstrated that the presence of comorbid headaches could be an important factor in determining whether riboflavin reduces headache frequency [26]. 

A recent retrospective study showed the efficacy of 100 and 200 mg of riboflavin in 42 patients (aged 6–18 years old): 38 with migraines, two with new, daily, persistent headaches, and two with chronic post-traumatic headaches [28]. There was a significant decrease in the number of days with headaches (frequency) after two to four months (11.07 ± 10.52 days) compared to baseline (21.90 ± 9.85 days). The mean headache intensity decreased from 8.85 ± 6.41 to 2.30 ± 2.51 (on a scale of 0 to 10; *p* < 0.001), and the headache duration significantly decreased (from 18.23 ± 17.07 to 10.18 ± 10.49 h; *p* < 0.001). A positive correlation was observed between the efficacy of riboflavin and reduced use of acute medications (rs ¼ = 0.304; *p* ¼ = 0.05) [28].

#### 3.1.3. Adverse Effects 

Studies using low doses of the riboflavin supplement did not report adverse events. In contrast, studies utilizing high doses reported orange discoloration of the urine, diarrhea, vomiting, and an increase in appetite without weight gain [22,24,27]. No studies documented any severe adverse events.

#### 3.1.4. Summary

Riboflavin can exert pharmacological effects even at low doses and may be useful in the prevention of migraine in children and adolescents. A prior pharmacokinetic study conducted among adults did not find significant differences in blood concentrations, maximal serum concentrations, or areas under the curve with 20, 40, and 60 mg of riboflavin supplementation [28]. As small doses of riboflavin (25 or 100 mg) have been shown to have pharmacological effects even in adults [46,47], similar effects are likely to occur in pediatric patients. However, the following factors have been reported to likely inhibit the effects of riboflavin: the presence of comorbid headaches, male sex, and age under 12 years [26,27]. However, there is no solid evidence, and further studies with larger numbers of patients and longer observational periods are required to assess the effectiveness of riboflavin-based prophylaxis for pediatric migraines. Riboflavin was not shown to have serious side effects and, therefore, it may be used as the first option when attempting to prevent pediatric migraines. 

### 3.2. Coenzyme Q10 

Coenzyme Q10 is required for cellular energy production, owing to its role as an electron carrier in the mitochondrial electron transport chain. Mitochondrial energy is depleted in migraines [15,16,17]; therefore, supplementation with coenzyme Q10 could help prevent migraines via the restoration of mitochondrial energy stores. Current data indicate that in migraine patients, the ingestion of coenzyme Q10 supplements leads to a reduction in the levels of calcitonin gene-related peptide (CGRP), which has been postulated to play an integral role in the pathophysiology of migraine [48]. Two studies, conducted by the same research group, have investigated the characteristics of coenzyme Q10 and its ability to prevent migraine [29,30].

In an open-label, uncontrolled study involving pediatric patients with migraines, a favorable response to coenzyme Q10 was observed among those with low coenzyme Q10 levels at baseline [29]. In a study, 252 children and adolescents with frequent headaches and deficiency in the serum levels of coenzyme Q10 were treated with 1–3 mg/kg of coenzyme Q10 supplements for a mean duration of approximately three months. The headache frequency was significantly reduced from 19.2 ± 10.0 to 12.5 ± 10.8 (*p* < 0.001), and the headache disability scores, assessed with pediatric migraine disability assessment (PedMIDAS), improved from 47.4 ± 50.6 to 22.8 ± 30.6 (*p* < 0.001). Further, 46% of the patients exhibited a 50% or greater reduction in headache frequency with increased serum concentration levels of coenzyme Q10 [29]. 

Subsequently, the same research group performed a crossover RCT to compare the effects of daily supplementation with 100 mg of coenzyme Q10 and a placebo for four months in children and adolescents with migraines. The patients treated with coenzyme Q10 showed a significant improvement in headache frequency only during the early stages of treatment, within four weeks of initiation. After the initial stage, no statistically significant difference was detected between the treated and control groups; both groups exhibited improvements in response to multidisciplinary standardized treatment, regardless of supplementation with coenzyme Q10 or the placebo. The planned sample size was 120 participants, of which 58 dropped out. The authors speculated that the high dropout rate was due to the rapid improvement associated with the coenzyme Q10 treatment. As a result, the subjects and their parents no longer felt that it was necessary to continue the treatment [30]. The lack of significant differences in the effects of coenzyme Q10 over the medium to long term might be due to the small sample size. Nevertheless, as the authors pointed out, multidisciplinary treatment and the placebo effect may have also influenced the results of the studies [30].

#### 3.2.1. Adverse Effects 

Although adverse effects of coenzyme Q10 treatment (anorexia, dyspepsia, nausea, and diarrhea) have been previously described [49], no serious adverse events have been reported in either of the included studies [29,30]. 

#### 3.2.2. Summary 

Despite coenzyme Q10 treatment not being associated with any serious adverse events, there is insufficient evidence to draw definitive conclusions on its efficacy.

### 3.3. Magnesium

Magnesium plays a key role in a wide variety of physiological and biochemical processes in the brain that are considered to be linked to migraine pathogenesis [14]. A growing body of evidence suggests that magnesium deficiency could contribute to an increased susceptibility to migraine development [50,51,52]. Magnesium has effects on the N-methyl-D-aspartate glutamate receptor blockade; glutamate and nitric oxide (NO) synthesis, release, and activity; serotonin receptor affinity and activity; and endogenous hormone regulation. Through these actions, magnesium is thought to regulate various vascular and neuronal mechanisms [14]. Two studies have evaluated the use of magnesium for pediatric and adolescent migraine prophylaxis [31,32].

A double-blind RCT on the efficacy of oral magnesium in the prevention of pediatric migraine included 118 children and adolescents with migraine [31]. The participants were randomized to receive either 9 mg/kg of elemental magnesium or a placebo, daily for 12 to 16 weeks. In a regression model, a significant downward trend in headache frequency was detected over time in the magnesium group compared to the placebo group. However, the time-by-treatment interaction term in the model was not significant, indicating that the slopes of the trends did not significantly differ between the groups; nevertheless, headache severity was significantly improved in the magnesium group compared to the placebo group [31]. 

A single-blind, non-randomized study included 160 children and adolescents with migraines assigned to four groups that underwent treatment with acetaminophen or ibuprofen, with or without magnesium. In this study, the preventive effects of magnesium, as well as the ameliorative effects of the acute analgesics (acetaminophen and ibuprofen), were examined and compared. Magnesium pretreatment induced a significant decrease in pain intensity (*p*  <  0.01) (without a time-dependent correlation in either the acetaminophen- or ibuprofen-treated children), and also significantly reduced (*p*  <  0.01) the time required for the onset of pain relief during acetaminophen (but not during ibuprofen) treatment (*p*  <  0.01). Magnesium pretreatment significantly reduced pain frequency in both the acetaminophen and ibuprofen groups (*p*  <  0.01) [32].

#### 3.3.1. Adverse Effects 

These studies consistently reported the presence of minor gastrointestinal problems, such as diarrhea and soft stools. No major adverse events were associated with magnesium supplementation.

#### 3.3.2. Summary 

The aforementioned RCT did not definitively ascertain the superiority of oral magnesium oxide over the placebo in the prevention of frequent migraine headaches in children [31]. Further research should be performed to assess the differential efficacy in comparison with the baseline magnesium levels. This would allow patients with magnesium deficiency to benefit further from magnesium supplementation and increased dietary intake.

### 3.4. Melatonin

Melatonin is a hormone released by the pineal gland into the bloodstream and it interacts with multiple organ systems [53]. Melatonin is critical for the functioning of human circadian rhythm and sleep and is suggested to play a role in the pathophysiology of migraines by activating melatonin receptors in the hypothalamus [54,55]. Clinical studies in migraine patients have reported lower levels of urinary melatonin metabolite on migraine days compared to non-migraine days in adults [56,57], but not in pediatric or adolescent patients [58]. While basic research studies have indicated that melatonin produces anti-inflammatory effects [59,60], a recent study reported a decrease in CGRP mRNA, NO, and nitrate levels upon the addition of melatonin to cultured human blood cells [61].

Two RCTs [33,34] and two open-label studies have investigated the role of melatonin in the prevention of migraine, among which three reported that melatonin was effective [34,35,36]. First, a three-month, open-label trial of melatonin treatment (0.3 mg/kg/day), using a small sample size of 22 children (14 with migraines and eight with tension-type headaches), demonstrated a reduction in the frequency of headaches. Ten of the 14 migraine patients exhibited a decrease of more than 50% in the frequency of headache attacks compared to baseline, and three patients reported having no headache attacks. Significant reductions were observed in headache frequency (from baseline until a three-month follow-up), headache frequency (from 12.3 ± 8.9 to 5.7 ± 6.7; *p* < 0.001), and headache duration (from 13.4 ± 15.9 to 9.7 ± 14.5 h; *p* < 0.001). Due to excessive daytime sleepiness, one subject dropped out after receiving melatonin for one month [35].

A recent open-label trial tested the efficacy of melatonin treatment for three months (0.3 mg/kg/day) in children with migraines and reported a significant decrease in the frequency and duration of headache attacks and disability levels [36]. The monthly frequency, severity scores, and duration of headaches were reduced from 15.63 ± 7.64 to 7.07 ± 4.42, 6.20 ± 1.67 to 3.55 ± 2.11, and 2.26 ± 1.34 to 1.11 ± 0.55 h, respectively. The PedMIDAS score decreased from 31.72 ± 8.82 to 17.78 ± 10.64 (*p* < 0.05). Further, 23.3% (*n* = 14) of the children exhibited clinical adverse events, including sleepiness (*n* = 7), vomiting (*n* = 4), mild hypotension (*n* = 2), and constipation (*n* = 1). Excessive daytime sleepiness that was considered a serious side effect was seen in three children; these cases required the discontinuation of melatonin [36].

#### 3.4.1. Positive Studies 

In a parallel, single-blinded RCT, the efficacy of amitriptyline (1 mg/kg/day) was compared to that of melatonin (0.3 mg/kg/day) in a population of 80 patients between 5 and 15 years with a history of migraine. A reduction of more than 50% in monthly headache frequency was observed in 82.5% and 62.5% of the patients in the amitriptyline and melatonin groups, respectively.

With melatonin use, the severity scores, duration, and PedMIDAS scores of the headaches were reduced from 6.05 ± 1.63 to 4.03 ± 1.54, 2.06 ± 1.18 to 1.41 ± 0.41 h, and 33.13 ± 9.17 to 23.38 ± 9.51, respectively. Both drugs were significantly effective in reducing the monthly frequency, severity, duration, and associated disability levels of the headaches. Amitriptyline was significantly more effective in improving all of these indicators. Daytime sleepiness was observed in 7.5% of the participants in the melatonin group [34].

In a similar RCT in children that compared amitriptyline (1 mg/kg/day) and melatonin (0.3 mg/kg/day), a total of 90 children with migraine (45 in each group)—aged between 5 and 15 years—were enrolled. The frequency of headaches along with their severity scores, duration, and PedMIDAS scores were noted monthly and compared after three months of treatment with studied drugs. At the end of the treatment period when both the groups were compared, the response, monthly frequency of headaches, severity, duration, and PedMIDAS scores were all significantly improved in the amitriptyline group (*p* < 0.05) [37].

#### 3.4.2. Negative Studies

In a home-based RCT, 31 participants were recruited at a clinic via flyers, social media, print advertisements, and letters to parents. Headache frequency was observed to be lower in the melatonin group compared to the placebo group during the final four weeks of treatment (the primary outcome measure). However, the difference was not significant; this may be attributed to the low number of participants. The mean number of migraine days was lower in the melatonin group than in the placebo group during the final four weeks of treatment. Nevertheless, this also did not reach statistical significance (mean (standard error) days, 3.6 (0.9) vs. 4.9 (1.7); difference, −1.3; 95% CI for difference, −5.1 to 2.6). The adjusted mean (standard error) number of migraine days was 3.1 (1.3) in the melatonin group and 5.4 (1.4) in the placebo group (difference, −2.3; 95% CI for difference, −6.3 to 1.8). Notably, despite the small number of participants, the study completion rate was quite favorable (89%) due to the use of remote devices, such as web-based electronic headache diaries and wearable devices for the recording of physiological data [33]. Research designs involving the use of remote devices might be important methods for future research of pediatric migraine.

#### 3.4.3. Adverse Effects 

Sleepiness, vomiting, and mild hypotension were observed [36]; however, no serious adverse events occurred.

#### 3.4.4. Summary 

There is a limited—but growing—number of available studies on the effectiveness of melatonin for migraine prophylaxis, most of which are seemingly valid. Melatonin has been demonstrated to be inferior to amitriptyline in the prevention of migraine in the aforementioned studies [34,37]. A recent systematic review and meta-analysis also showed that the efficacy of melatonin for migraine has not been established in adults [62]. While a recently conducted RCT reported that melatonin was effective for acute migraine attacks [63]. Melatonin is vital for sleep function; further, the link between migraine and sleep disturbance is well known, and a bidirectional relationship has also been reported during childhood [64,65]. Reduced melatonin levels during the night prior to a migraine attack were observed in some pediatric patients, and supplementation with melatonin may be particularly beneficial in these patients [58]. Melatonin might be a safe integrative prophylactic treatment for children with primary headaches; however, further evaluation is required to confirm its effects.

### 3.5. PUFAs

Long-chain n-3 PUFAs, including eicosapentaenoic acid (EPA) and docosahexaenoic acid (DHA), are essential dietary components that are not synthesized by humans. In addition to the well-established anti-vasopressor effects of PUFAs [66], they are required for the optimal functioning of the nervous system [67] and have known anti-inflammatory properties [68,69,70].

Two RCTs with small sample sizes have evaluated the effectiveness of PUFAs for the prevention of migraine in adolescents. In a crossover RCT, 27 adolescents with chronic migraines were randomized to receive either a fish oil compound containing EPA, DHA, and tocopherol or a placebo capsule containing olive oil for two months. The participants exhibited significant reductions in headache frequencies with both the fish oil and placebo treatments; no significant difference was observed. The lack of a significant difference in this study may be attributed to the small number of cases and a strong placebo effect. Indeed, this study raised questions about the suitability of olive oil as a placebo [38]. 

The second RCT utilized a double-blind, parallel-group design, and allocated 25 adolescents with migraines to receive either a daily treatment of 20 mg/kg of sodium valproate and a fish oil compound containing EPA and DHA or 20 mg/kg sodium valproate and a placebo compound for two months. Both groups exhibited significant reductions in headache frequencies and PedMIDAS scores; however, there were no statistically significant differences between the groups. One patient in the PUFA group withdrew from the study due to nausea [39]. However, there is insufficient evidence that PUFAs can potentially reduce the frequency and intensity of migraine attacks; therefore, a definitive conclusion cannot be drawn.

### 3.6. Combination Therapy 

Although we have described the CIMs that appear to have the greatest potential effectiveness for preventing pediatric migraine, the use of several other agents has been evaluated as combination therapy. We have briefly described such treatments below.

#### 3.6.1. Feverfew (*Tanacetum parthenium* (L.)) 

Feverfew, also known as *Tanacetum parthenium*, is a medicinal herb that has been used for a variety of ailments for centuries. It has been reported to influence processes that are implicated in migraine pathogenesis [71] and exhibit antimigraine actions, such as the inhibition of NO synthesis and cytokine production and CGRP induction, and the subsequent promotion of serotonin release from platelets [72].

A recent RCT assessing the use of combination therapy (feverfew, coenzyme Q10, riboflavin, and magnesium) demonstrated favorable outcomes [40]. Although the treatment was interrupted in 4.4% of the patients due to gastrointestinal symptoms (nausea and diarrhea), it was not reported whether these side effects were due to the use of feverfew. Furthermore, the use of feverfew alone for the prevention of migraine in children has not been investigated to date.

#### 3.6.2. Vitamin D

Vitamin D acts as a neurosteroid in autocrine and paracrine signaling and is thought to play a role in brain development, neurotransmission, synaptic plasticity, cell death prevention, and amyloid clearance [73]. However, it is unclear whether vitamin D has a specific role in migraine prevention. While a prospective study demonstrated that vitamin D administered in addition to amitriptyline reduced the number of migraine attacks, safety outcomes were not reported [41]. In a recent single-blinded, randomized, clinical trial of children with migraine headaches (5–15 years old), the combination of topiramate and vitamin D3 showed a significant effect on the reduction of monthly headache frequency (6.12 ± 1.26 vs. 9.87 ± 2.44, *p* = 0.01) and disability score (19.24 ± 6.32 vs. 22.11 ± 7.91, *p* = 0.02) compared with topiramate alone [42]. Additionally, in a current meta-analysis of serum levels of 25-hydroxyvitamin D3 in migraine patients, preliminary evidence has identified low levels in both adults and children [74]. The potential efficacy of vitamin D suggests that CIM with an attractive safety profile could be used in combination regimens.

#### 3.6.3. Ginkgolide B

Ginkgolide B, an herbal constituent extract from *Ginkgo biloba* tree leaves, is a natural modulator of the action of glutamate in the central nervous system [75]. Based on the results of open-label studies on migraine prevention in children, ginkgolide B (in combination with a variety of other CIMs) appears to produce favorable results without any adverse events [43,44,45]. 

#### 3.6.4. Summary 

The use of various combinations of feverfew, vitamin D, and ginkgolide B for pediatric migraine prevention appears to be effective without serious adverse effects. However, due to the lack of stand-alone studies, the individual effect of each integrative agent cannot be determined. 

## 4. Conclusions

In this review, we evaluated the current state of evidence regarding the use of CIMs for pediatric and adolescent migraine. As there was a lack of high-quality evidence, with most studies limited by small sample sizes, we were unable to definitively identify an optimal CAM for migraine prophylaxis in children and adolescents. Although the level of evidence for riboflavin was seen to be relatively high, there is a need for additional studies to evaluate the effectiveness of CIMs. 

The effectiveness of placebos for the prevention of childhood migraine has been reported to range from 30–60% [6,76]; thus, this may be an obstacle for the accurate evaluation of CAM effectiveness. Pediatricians should take advantage of the available knowledge on placebo and nocebo responses to expand therapeutic outcomes [77]. In addition, pediatricians should be aware of how important it is for patients and their parents to realize that they are receiving optimal care from their doctors. It has been well documented—but often overlooked—that both environmental factors and endogenous physiological mechanisms play central roles in controlling pediatric migraine attacks [78,79]. To avoid interfering with a child’s development, it is necessary to identify and select effective CIMs and dosages that result in minimal side effects. Simply administrating CIMs is not enough; the affected children and their parents/guardians must maintain a balanced lifestyle. This would lead to a healthy child and family [79]. 

Taking into consideration the advantages of these psychobiological therapeutic responses, it should be explored whether CIMs might be useful in targeting specific aspects of the migraine pathophysiology. Currently, the body of evidence supporting the use of CIMs is undoubtedly small, but it is slowly and surely growing. While CIMs may be potentially used as a first-line treatment for pediatric migraine prophylaxis, additional high-quality studies are required to confirm their effectiveness and safety.

In the current state of conclusion, it is premature to affirm the effectiveness of recommending integrative medicine treatments to patients. However, it will be useful to make some recommendations and highlight the need for proceeding with caution due to lack of evidence.

## Figures and Tables

**Table 1 jcm-10-00138-t001:** Summary of studies on complementary and integrative medicines for pediatric migraine.

Diet	References: Study Design	*N*/Patients	Intervention	Comparison	Outcomes	Adverse Effects
Riboflavin	Athaillah et al., 2012 [22]: RCT	98 adolescents with migraine	Riboflavin (400 mg daily) for three months (*n* = 50)	Placebo for three months (*n* = 48)	Headache frequency was decreased from the 1st month (6.4 episodes/month) to the 2nd month (3.9) to the 3rd month (3.7 per month) and duration decreased (*p* = 0.012 and *p* = 0.001, respectively) vs. placebo. Disability, as measured by the PedMIDAS was also decreased (34.3 to 26.1; *p* = 0.001).	Polyuria (*n* = 18), and diarrhea (*n* = 12)
	Talebian et al., 2018 [23]: RCT	90 children and adolescents with migraine	Riboflavin (200 mg or 400 mg daily) for 3 months (*n* = 30, and 30, respectively)	Placebo for three months (*n* = 30)	The Riboflavin 400 mg group had a greater reduction in the headache frequency (9.3 to 2.9 episodes/month) and duration (7.9 to 3.9 episodes/month) as compared to placebo (*p* = 0.00 and *p* = 0.00, respectively).	None
	MacLennan et al., 2008 [24]: RCT	48 children and adolescents with migraine	Riboflavin (200 mg daily) for 12 weeks (*n* = 27)	Placebo for 12 weeks (*n* = 21)	No difference between the groups in terms of the proportion of participants with 50% or greater reduction of migraine frequency (44.4% of the riboflavin vs. 66.6% of the placebo group, *p* = 0.125)	Change in urine color (*n* = 1)
	Bruijn et al., 2010 [25]: Crossover RCT	42 children with migraine	Riboflavin (50mg daily) for four months (*n* = 20)	Placebo for four months (*n* = 22)	No difference between groups in terms of change in migraine frequency (*p* = 0.44); the riboflavin group had a greater reduction in the frequency of tension-type headaches as compared to placebo (*p* = 0.04)	None
	Yamanaka et al., 2020 [26]: Retrospective observational study	68 children and adolescents with migraine	Riboflavin (10 or 40 mg daily) for three months (*n* = 13, and 55, respectively)	N/A	Significant overall reduction in the median frequency of headache episodes from the baseline to three months (median, 5.2 vs. 4.0 respectively, *p* = 0.00) was shown.	None
	Condo et al., 2009 [27]: Retrospective chart review	41 children and adolescents with a variety of headache disorders	Riboflavin (200 mg or 400 mg daily) for three, four or six months	N/A	Significant reduction in headache frequency after treatment for three or four months (21.7_13.7 vs. 13.2_11.8, *p* < 0.01), which was not sustained at six months (19.3_13.4 vs. 11.4_9.6, *p* > 0.05)	Vomiting (*n* = 1), increased appetite without weight gain (*n* = 1), and temporary yellow-orange coloration of urine (some patients)
	Das et al., 2020 [28]: Retrospective observational study	42 children and adolescents with a variety of headache disorders	Riboflavin (100 and 200 mg for children weighing 20 to 40 kg and greater than 40 kg, respectively)	N/A	There was a significant decrease in the number of headache days (frequency) after 2 to 4 months (11.07 ± 10.52 days) compared to baseline (21.90 ± 9.85 days). Mean headache intensity decreased from 8.85 ± 6.41 to 2.30 ± 2.51 (*p* < 0.001 on a scale of 0 to 10), and headache duration also decreased significantly from 18.23 ± 17.07 h to 10.18 ± 10.49 h; *p* < 0.001).	None
Coenzyme Q10	Hershey et al., 2007 [29]: Prospective open-labeled study	252 children and adolescents with migraine	Coenzyme Q10 (1–3 mg/kg daily) for average of three months	N/A	The headache frequency improved from 19.2 ± 9.8 days per month to 12.5 ± 10.8 days per month (*p* < 0.001). PedMIDAS score improved from 47.4 ± 50.6 to 22.8 ± 30.6 (*p* < 0.001) (grade improved from 2.6 ± 1.2 to 1.9 ± 1.0 (*p* < 0.001). Headache duration decreased from 11.7 ± 18.1 to 5.7 ± 9.1 h (*p* < 0.001). The 46% of patients achieving a 50% or greater reduction in headache frequency.	N/D
	Slater et al., 2011 [30]: Crossover, add-on RCT	120 children and adolescents with migraine	Coenzyme Q10 (100 mg daily) for four months (n = 60)	Placebo for four months (*n* = 60)	Significant reduction in headache frequency in both groups, but repeated measures ANOVA failed to show a time condition interaction (*p* > 0.05). No significant differences between both groups in migraine severity and duration.	N/D
Magnesium	Wang et al., 2003 [31]: RCT	118 children and adolescents with headaches suggestive of migraine	Magnesium oxide containing 9 mg/kg of elemental magnesium for four months (*n* = 58)	Placebo for four months (*n* = 60)	Both groups had a downward trend in headache days, with only the magnesium group sustaining the trend past six weeks; no significant difference between groups after regression analyses (*p* = 0.88)	Diarrhea or soft stools (*n* = 11)
	Gallelli et al., 2014 [32]: Single-blind RCT	160 children and adolescents with migraine	Magnesium (400 mg daily) and acetaminophen (15 mg/kg) or ibuprofen (10 mg/kg) to be taken with acute migraine episodes for 18 months (*n* = 40 and 40, respectively)	Acetaminophen (15 mg/kg) or ibuprofen (10 mg/kg) to be taken with acute migraine episodes for 18 months (*n* = 40 and 40, respectively)	Treatment with magnesium reduced pain intensity acutely when combined with acetaminophen or ibuprofen (*p* < 0.01) and resulted in a reduction of migraine frequency (*p* < 0.01)	None
Melatonin	Gelfand et al., 2017 [33]: Small RCT (home-based trials using social media)	31 children and adolescents with migraine	Melatonin 3 mg daily for three months (*n* = 18)	Placebo for three months (*n* = 13)	Mean migraine days was lower in the melatonin group vs the placebo group in the final 4 weeks of treatment (the primary outcome measure) but was not statistically significant..	Daytime tiredness (*n* = 2), low iron on blood work (*n* = 1), and vomiting (*n* = 1)
	Fallah et al., 2018 [34]: Parallel, single-blinded randomized clinical study	80 children and adolescents with migraine	Melatonin (0.3 mg/kg, max 6 mg daily) for three months (*n* = 40)	Amitriptyline (1 mg/kg, max 50 mg daily) for three months (*n* = 40)	Significant reduction in headache frequency after treatment for three months was seen in 62.5%. The severity scores, duration, and PedMIDAS scores of the headaches reduced from 6.05 ± 1.63 to 4.03 ± 1.54, 2.06 ± 1.18 to 1.41 ± 0.41 h, and 33.13 ± 9.17 to 23.38 ± 9.51, respectively. But amitriptyline was significantly more effective in improving all of these indicators.	Excessive daytime sleepiness (*n* = 2)
	Mirano et al., 2008 [35]: Prospective open-label trial study	14 children and adolescents with migraine	Melatonin (3 mg/day daily) for three months	N/A	10 patients reported that the headache attacks had decreased by more than 50% in respect to baseline.	Excessive daytime sleepiness (*n* = 1)
	Fallah et al., 2015 [36]: Prospective open-label trial study	38 children and adolescents with migraine	Melatonin (0.3 mg/kg daily) for three months	N/A	Monthly frequency, severity and duration of headache reduced from 15.63 ± 7.64 to 7.07 ± 4.42 attacks, from 6.20 ± 1.67 to 3.55 ± 2.11 scores, and from 2.26 ± 1.34 to 1.11 ± 0.55 h, respectively.	Sleepiness (*n* = 7): excessive daytime sleepiness (*n* = 3), vomiting (*n* = 4), mild hypotension (*n* = 2), and constipation (*n* = 1)
	Shahnawaz et al., 2019 [37]: Parallel, single-blinded randomized clinical study	45 children with migraine	0.3 mg/kg (maximum 6 mg) at bedtime	Amitriptyline (1 mg/kg, maximum 50 mg) at bedtime (*n* = 45)	Significant better in amitriptyline group (*p* value < 0.05) in monthly frequency of headache, severity and duration, and headache disability PedMIDAS.	Side effects (35.6%), No serious adverse event.
Polyunsaturated fatty acids	Harel et al., 2002 [38]: Crossover RCT	27 adolescents with migraine	Marine *n*-3 ethyl ester concentrate (two capsules daily) for two months (*n* = 14)	Placebo for two months (*n* = 13)	Significant reduction in headache frequency during Marine n-3 ethyl ester concentrate treatment and during placebo treatment but no significant difference between these treatments.	None
	Fayyazi et al., 2016 [39]: Prospective open-labeled study	25 children and adolescents with migraine	Omega-3 capsule (containing 1 g of fish oil) and valproate (20 mg/kg, max 200 mg) daily for two months (*n* = 12)	Placebo with valproate (20 mg/kg, max 200 mg daily) for two months (*n* = 13)	Significant reduction in headache frequency and PedMIDAS scores in case group and control group, but there were no statistically significant differences observed between the groups.	Nausea (*n* = 1)
Feverfew	Moscano et al., 2019 [40]: Observational multicenter study	71 children and adolescents with migraine	Partena® tablets (Mg^2+^, CoQ10, VitB2, Feverfew, Parthenolides, Andrographis paniculate) daily for four months	N/A	In the assessment of both headache frequency and intensity, a significant effect of treatment and maintenance of it was found.	N/D
Vitamin D	Cayir et al., 2014 [41]: Prospective open-labeled study	53 children and adolescents with migraine	Vitamin D supplementation (400 or 800 or 5000 IU) and Amitriptyline (1 mg/kg) daily for six months (*n* = 40)	Only amitriptyline (1 mg/kg daily) for six months (*n* = 13)	There was a significant decrease in migraine attacks in the groups receiving vitamin D compared with the group receiving amitriptyline alone.	N/D
	Fallah et al., 2020 [42]: In a single-blinded, randomized, clinical trial	57 children and adolescents with migraine	2 mg/kg/day of topiramate (*n* = 28) or 2 mg/kg/day of topiramate plus one 500,000 IU vitamin D3 pearl (*n* = 29) weekly for two consecutive months	N/A	The combination group of topiramate and vitamin D3 showed a significant effect on the reduction of monthly headache frequency (6.12 ± 1.26 vs. 9.87 ± 2.44, *p* = 0.01) and disability score (19.24 ± 6.32 vs. 22.11 ± 7.91, *p* = 0.02) compared with the topiramate alone group	Daily sleepiness (*n* = 2), constipation (*n* = 2), and anorexia (*n* = 1)
Ginkgolide B	Usai et al., 2010 [43]: Prospective open-label study	24 children and adolescents with migraine	Ginkgolide B, Coenzyme Q10, Vitamin B2, and Magnesium daily for three months	N/A	Starting with a mean baseline of 7.4 ± 5 attacks, clinical improvement was significant: the mean number of days of headache per month decreased to 2.2 ± 2.8 (*p* = 0.0015), with a decrease of number of analgesics used for the attacks from 5.9 ± 5.3 to 1.5 ± 2.2 (*p* = 0.013).	N/D
	Esposito et al., 2011 [44]: Prospective open-label study	119 children and adolescents with migraine	Ginkgolide B, Coenzyme Q10, Riboflavin, and Magnesium daily for three months	N/A	The mean frequency per month of migraine was significantly decreased (9.71 ± 4.33 vs. 4.53 ± 3.96 attacks; *p* < 0.001).	N/D
	Esposito et al., 2012 [45]: Prospective open-label study	374 children and adolescents with migraine	Ginkgolide B, Coenzyme Q10, Riboflavin, and Magnesium daily for six months (*n* = 187)	L-tryptophan 5-hydroxytryptophan (from Griffonia simplicifolia), Vitamin PP, and Vitamin B6 daily for six months (*n* = 187)	Both preparations reduced all outcome measures, but reductions in headache frequency, duration and intensity, PedMIDAS score and behavioral reactions to headache were significantly greater in the ginkgolide B group. The ginkgolide B preparation was significantly more effective in the medium-term (six months).	None

RCT: randomized controlled trial; MIDAS: Migraine Disability Assessment; N/A: not available. N/D: not described.

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
