# Peer review of "Complementary and Integrative Medicines as Prophylactic Agents for Pediatric Migraine: A Narrative Literature Review"

_jcm, 2021, doi:10.3390/jcm10010138_

Round 1
Reviewer 1 Report
The article ‘Complementary and Alternative Medicines as Prophylactic Agents for Pediatric Migraine: A Narrative Literature Review’ aimed to summarize the published evidence for the use of CAMs in the prevention of pediatric migraine. This is a very important and interesting subject. However the present paper requires major modifications, as well as English editing.
As noticed by the authors, placebo effect is substantial among children with migraine (Powers et al., 2017. NEJM). The translation of the results from uncontrolled trials should always take into consideration this large placebo effect, since it may account partly if not entirely for the efficacy of an intervention. Statistically significant within group differences should be interpreted with caution as they cannot be evaluated on their own. In order to capitalize within group differences, the authors could compare them with the outcome-specific placebo responses from other published studies. Similar migraine populations should be used if available (regarding age, sex, race, subtype of migraine e.g. episodic or chronic, etc). For even more accurate conclusions, the highest recorded outcome-specific placebo responses could be utilized. Otherwise, the independent interpretation of uncontrolled studies carries a risk of misinterpreting a placebo effect as evidence of a prophylactic effect against migraine.
For instance, the authors report the accumulation of favorable evidence for the use of melatonin in migraine prophylaxis based on 1) an RCT that reported a similar efficiency between melatonin and placebo 2) an RCT that reported an inferior efficiency of melatonin compared to amitriptyline 3) the significant within group efficacy-results of RCTs (n=2) and non-RCTs (n=2). In fact, these results are compatible with an overall efficacy of melatonin similar to placebo and inferior to amitriptyline. Within group differences cannot be capitalized without proper comparisons. Taking a more careful look, we observe that monthly headache frequencies during the last month of treatment (assessed outcome) were higher in all of the uncontrolled studies for melatonin (Fallah et al., 2015; Milano et al., 2008), compared to the monthly headache frequency provided by the only available placebo group (Gelfand et al. 2017). Therefore, the conclusion that the use melatonin could be of potential benefit in the prevention of pediatric migraine is completely unfounded.
Aside from that, according to the authors, published systematic reviews and meta-analyses were retrieved along with relevant RCTs and observational studies. Considering the present paper as an umbrella review for multiple interventions, the inclusion of systematic reviews and meta-analyses is warranted. Nonetheless, literature search failed to retrieve all of the relevant articles. For example, the recently published systematic review of Liampas et al. (2020, Headache) involved an additional RCT for melatonin in pediatric migraine (Shahnawaz et al., 2019, Med Forum). Therefore, the said systematic review and by extension the said RCT should be included in the present paper (according to the reported literature algorithm). A reassessment and probably a repetition of the literature search are probably warranted (considering the outdated final literature search as well).
Finally, reporting should be more concise. For each study, results should be reported and translated per migraine subgroup (e.g. migraine with aura – migraine without aura, episodic migraine – chronic migraine). Several compounds may be useful only for a specific subtype of migraine. Of course, retrieved studies may present only pooled results (e.g. episodic with chronic migraine), but these reporting deficiencies should be clearly stated by the authors.
Author Response
To the Reviewer 1
We appreciate the time and effort you have dedicated to provide this insightful feedback. We have revised our manuscript in accordance with your comments, as much as possible. However, we have limited time and may not have been able to make all the corrections that you mentioned. All suggested revisions, as well as additional ones to improve the language of the manuscript, are indicated in underlined text. We hope that, with these revisions, our manuscript will be suitable for publication.
Responses to Reviewer 1’s comments
The article ‘Complementary and Alternative Medicines as Prophylactic Agents for Pediatric Migraine: A Narrative Literature Review’ aimed to summarize the published evidence for the use of CAMs in the prevention of pediatric migraine. This is a very important and interesting subject. However, the present paper requires major modifications, as well as English editing.
Comment 1
As noticed by the authors, placebo effect is substantial among children with migraine (Powers et al., 2017. NEJM). The translation of the results from uncontrolled trials should always take into consideration this large placebo effect, since it may account partly if not entirely for the efficacy of an intervention. Statistically significant within group differences should be interpreted with caution as they cannot be evaluated on their own. In order to capitalize within group differences, the authors could compare them with the outcome-specific placebo responses from other published studies. Similar migraine populations should be used if available (regarding age, sex, race, subtype of migraine e.g. episodic or chronic, etc.). For even more accurate conclusions, the highest recorded outcome-specific placebo responses could be utilized. Otherwise, the independent interpretation of uncontrolled studies carries a risk of misinterpreting a placebo effect as evidence of a prophylactic effect against migraine.For instance, the authors report the accumulation of favorable evidence for the use of melatonin in migraine prophylaxis based on 1) an RCT that reported a similar efficiency between melatonin and placebo 2) an RCT that reported an inferior efficiency of melatonin compared to amitriptyline 3) the significant within group efficacy-results of RCTs (n=2) and non-RCTs (n=2). In fact, these results are compatible with an overall efficacy of melatonin similar to placebo and inferior to amitriptyline. Within group differences cannot be capitalized without proper comparisons. Taking a more careful look, we observe that monthly headache frequencies during the last month of treatment (assessed outcome) were higher in all of the uncontrolled studies for melatonin (Fallah et al., 2015; Milano et al., 2008), compared to the monthly headache frequency provided by the only available placebo group (Gelfand et al. 2017). Therefore, the conclusion that the use melatonin could be of potential benefit in the prevention of pediatric migraine is completely unfounded.
Response: We deeply appreciate this suggestion and completely agree.
Indeed, the placebo effect is high in the treatment of pediatric patients with migraine, thereby hindering proof of the drugs. Unfortunately, no evidence has been established for the drugs cited in this article.
Therefore, the contents of the text have been reexamined, and the following revision has been incorporated in the conclusion (page 14, lines 321–324).
In the current state of inference, it is indeed premature to affirm the effectively the recommendation of integrative medicine treatments to our patients. However, it will be useful to make certain recommendations while highlighting the need for a cautious approach due to lack of evidence.
Comment 2
Aside from that, according to the authors, published systematic reviews and meta-analyses were retrieved along with relevant RCTs and observational studies. Considering the present paper as an umbrella review for multiple interventions, the inclusion of systematic reviews and meta-analyses is warranted. Nonetheless, literature search failed to retrieve all of the relevant articles. For example, the recently published systematic review of Liampas et al. (2020, Headache) involved an additional RCT for melatonin in pediatric migraine (Shahnawaz et al., 2019, Med Forum). Therefore, the said systematic review and by extension the said RCT should be included in the present paper (according to the reported literature algorithm). A reassessment and probably a repetition of the literature search are probably warranted (considering the outdated final literature search as well).
Response: Thank you for your valuable feedback. We had overlooked this paper since we had searched using the PubMed database.
As you have mentioned, since it has been cited in other revisions of the book, I will cite it yet again. As for the other CIMs, I searched yet again but was unable to find any relevant study.
We have revised the following text (page, lines 210–216).
In similar RCTs in children regarding the comparison of amitriptyline (1 mg/kg/day) and melatonin (0.3 mg/kg/day), a total of 90 children with migraine (45 in each group)—aged between 5–15 years—were enrolled. The frequency of headaches along with their severity, duration, and pedMIDAS were noted monthly and compared after 3 months of treatment with the studied drugs. At the end of the treatment period when both groups were compared, good response, monthly frequency of headache, severity, duration, and headache disability pedMIDAS were all significantly improved in the amitriptyline group (p < 0.05).
Comment 3
Finally, reporting should be more concise. For each study, results should be reported and translated per migraine subgroup (e.g. migraine with aura, migraine without aura, episodic migraine, chronic migraine). Several compounds may be useful only for a specific subtype of migraine. Of course, retrieved studies may present only pooled results (e.g. episodic with chronic migraine), but these reporting deficiencies should be clearly stated by the authors.
Response: Thank you for your precise comment. Based on the feedback we received, we have revised the manuscript as much as possible. However, due to limited time, we may not have been able to revise based on all your suggestions. If needed, we will revise it again, and we would appreciate any further feedback.
 Thank you very much.
Reviewer 2 Report
Title: I would adjust the title of the paper to Complementary and Integrative Medicine as Prophylactic Agents for Pediatric Migraine: A Narrative Literature Review
The national center for integrative health has recommended to now use integrative medicine rather than alternative medicine.
Abstract:
- The following statement "Migraine headaches are common in children and adolescents, but no prophylaxis for migraine has been established" is incorrect based on pediatric migraine preventive guidelines - amitriptyline, topiramate, cyproheptadine and other have been approved.
- Would change alternative medicine to integrative medicine as recommended in title.
Introduction
- Would avoid using the term migraines, and use migraine instead.
- would add more about the epidemiology of pediatric migraine - how prevalent is it and gender predominance.
- Would Change CAM (complementary and alternative medicine) to CIM (complementary and alternative medicine)
The "summary of search section" should be divided into METHODS and RESULTS. The table provided should be placed in the results section.
I would add a study limitations and discussion sections.
Conclusion - In the current state of the conclusion, I do not understand if this study is helpful or not or whether I should recommend integrative medicine treatments to my patients or not. I think some recommendations should be made and then highlight that one should proceed with caution due to lack of evidence.
Author Response
To the Reviewer 2
We would like to thank you for carefully reviewing our manuscript and providing your valuable comments. We have revised our manuscript based on your comments. All the suggested revisions, as well as the additional ones to improve the language of the manuscript, are indicated by the underlined text. We hope that, with these revisions, our manuscript will now be suitable for publication.
Responses to Reviewer 2’s comments
Abstract:
- The following statement "Migraine headaches are common in children and adolescents, but no prophylaxis for migraine has been established" is incorrect based on pediatric migraine preventive guidelines - amitriptyline, topiramate, cyproheptadine and other have been approved.
Response: The following sentences in the abstract have been omitted.
Migraine headaches are common in children and adolescents, but no prophylaxis for migraine has been established.
- Would change alternative medicine to integrative medicine as recommended in title.
Response: Thank you for your valuable insight. Please note that all mentions of “alternative medicine” have been changed to integrative medicine. “CAMs” have also been changed to “CIMs.”
Introduction
- Would avoid using the term migraines, and use migraine instead.
Response: We have replaced the term “migraines” with “migraine.”
- would add more about the epidemiology of pediatric migraine - how prevalent is it and gender predominance.
Response: Thank you very much for your valuable constructive feedback. Please find our edited passage below (page 1, lines 32–37).
Migraine headaches are common, occurring in 3–5% of young children and up to 18% of adolescents {Lewis, 2004 #1759}. Migraine incidence appears to peak earlier in men than in women {Stewart, 2008 #1757}{Lipton, 2005 #1758}, and the sex ratio varies considerably with age; although this ratio is approximately equal before puberty{Lipton, 2005 #1758}{Merikangas, 2013 #1323}, across the lifespan, women are two to three times more likely to experience a migraine than men {Merikangas, 2013 #1323}.
- Would Change CAM (complementary and alternative medicine) to CIM (complementary and alternative medicine)
Response: “CAMs” have also been changed to “CIMs.”
The "summary of search section" should be divided into METHODS and RESULTS. The table provided should be placed in the results section.
Response: Thank you for your notable feedback. Please note that we have corrected this based on your instructions (page 1, lines 65–76).
- I would add a study limitations and discussion sections.
Response: Thank you very much for your constructive feedback. We have added the following concluding statements to the text per your recommendations: (page 1, lines 321–324)
In the current state of inference, it is indeed premature to affirm effectively the recommendation of integrative medicine treatments to our patients. However, it will be useful to make certain recommendations while highlight the need for a cautious approach due to lack of evidence.
Round 2
Reviewer 1 Report
I appreciate the work of the authors and the extensive English editing. However, once again I have to point out that the translation of the results is not in accordance with your findings. Please follow my recommendations below to avoid the extraction of any misleading conclusions. Moreover, the literature search was suboptimal. I noticed that the date of the literature search was not different in your revision (compared to your submission). There are several important recently published studies that have to be included. Below, I quote the relevant recent references that were not included in the present review. These studies should be included as they considerably enhance the strength of your findings.
Riboflavin: Among the investigated CIMs, riboflavin accumulated the greatest research interest in pediatric migraine. Your findings were indeed indicative of a potential beneficial effect of riboflavin, but additional RCTs are required. An uncontrolled trial investigating the use of riboflavin in children and adolescents was recently published and should be included (Das R, Qubty W. Retrospective Observational Study on Riboflavin Prophylaxis in Child and Adolescent Migraine. Pediatr Neurol. 2020 Sep 24;114:5-8. doi: 10.1016/j.pediatrneurol.2020.09.009. Epub ahead of print. PMID: 33189027.).
Coenzyme Q10: Summary: The authors of the included RCT conclude ‘that children and adolescents with migraine improved over time with multidisciplinary, standardized treatment regardless of supplementation with CoQ10 or placebo’. However you conclude that ‘evidence appears to support the efficacy of coenzyme Q10 in preventing migraine in children and adolescents’. Observational studies are of limited value in case an RCT has been published, therefore, you should base your conclusions on the results of the RCT. According to the results of the included RCT, Q10 is similarly efficacious to placebo, therefore, there is no evidence suggesting its preventive effect. Please reformulate your conclusion.
Magnesium: Summary: Wang et al. (2003) conclude that ‘This study does not unequivocally determine whether oral magnesium oxide is or is not superior to placebo in preventing frequent migrainous headache in children’. These conclusion was reached because the authors ‘were not able, however, to show that the slopes of the 2 lines were significantly different from each other’. On the other hand, Gallelli et al. (2013) performed a single blind study, in which physicians rather than participants were blinded: ‘Physicians who were blinded to the treatment assessed the overall clinical response during the study’. In view of the awareness of the participants, it is probable that the efficacy of magnesium is due to a placebo effect. Therefore, according to the findings of the first RCT magnesium is equally efficacious with placebo (the second RCT is of lesser quality in view of its open-label design participant-wise). Please reformulate your conclusion that ‘magnesium is effective in preventing migraine in children and adolescents’. There is no such evidence.
Melatonin: Summary: You mentioned that most retrieved studies for melatonin ‘have demonstrated favorable outcomes’. Again, melatonin seems to be as efficacious as placebo according to the presented findings. Please reformulate. Liampas et al. have demonstrated that melatonin is inferior to amitriptyline (meta-analysis) in the prevention of migraine (Liampas I, Siokas V, Brotis A, Vikelis M, Dardiotis E. Endogenous Melatonin Levels and Therapeutic Use of Exogenous Melatonin in Migraine: Systematic Review and Meta-Analysis. Headache. 2020 Jul;60(7):1273-1299. doi: 10.1111/head.13828. Epub 2020 Apr 30. PMID: 32352572.). This study should be included and the inferiority of melatonin to amitriptyline should be pointed out. An intriguing recent RCT found that melatonin could be even effective in acute attacks (Gelfand AA, Ross AC, Irwin SL, Greene KA, Qubty WF, Allen IE. Melatonin for Acute Treatment of Migraine in Children and Adolescents: A Pilot Randomized Trial. Headache. 2020 Sep;60(8):1712-1721. doi: 10.1111/head.13934. PMID: 32965037.). I recommend that this finding is at least mentioned by the authors because it may be potentially useful in the understanding of the pathogenetic mechanisms involved in pediatric migraine
PUFAs: Paragraph 3: According to the presented findings, existing evidence is suggestive that PUFAs are as effective as placebo, therefore, there is no evidence that ‘PUFAs may potentially reduce the frequency and intensity of headache attacks’. Please reformulate to reflect these findings accurately (that PUFAs have not yet proved useful in migraine prophylaxis).
Vitamin D: Fallah et al. recently published an article comparing the use of topiramate vs. topiramate + vitamin D3 in the prophylaxis of pediatric migraine (Fallah R, Sarraf Yazd S, Sohrevardi SM. Efficacy of Topiramate Alone and Topiramate Plus Vitamin D3 in the Prophylaxis of Pediatric Migraine: A Randomized Clinical Trial. Iran J Child Neurol. 2020 Fall;14(4):77-86. PMID: 33193786; PMCID: PMC7660031.). This study should be included and the fact that CIMs with attractive safety profiles could be used in combination regimens should be stressed. Moreover, Liampas et al. recently published a meta-analysis for the serum levels of 25(OH)D among patients with migraine and found lower 25(OH)D concentrations among adults and preliminary evidence suggesting that lower concentrations may be found in children too (one study that revealed lower concentrations and one study that found equal levels of 25(OH)D compared to healthy controls were included) (Liampas I, Siokas V, Brotis A, Dardiotis E. Vitamin D serum levels in patients with migraine: A meta-analysis. Rev Neurol (Paris). 2020 Sep;176(7-8):560-570. doi: 10.1016/j.neurol.2019.12.008. Epub 2020 Mar 30. PMID: 32241571.). This study should be included to support the evidence for the potential effectiveness of vitamin D.
Conclusions: Paragraph 1: ‘there is an urgent need for additional studies’, Please tone down your wording. Urgent is a very strong word.
Author Response
To the Reviewer 1
We appreciate the time and effort you have dedicated to providing this insightful feedback. We have revised our manuscript in accordance with your comments, as much as possible. However, due to time constraints, we were unable to address all the corrections you mentioned. All suggested revisions and additional changes made to improve the language of the manuscript are indicated as underlined text. We hope that, with these revisions, our manuscript will be suitable for publication.
Responses to Reviewer 1’s comments
Comment 1
I appreciate the work of the authors and the extensive English editing. However, once again I have to point out that the translation of the results is not in accordance with your findings. Please follow my recommendations below to avoid the extraction of any misleading conclusions. Moreover, the literature search was suboptimal. I noticed that the date of the literature search was not different in your revision (compared to your submission). There are several important recently published studies that have to be included. Below, I quote the relevant recent references that were not included in the present review. These studies should be included as they considerably enhance the strength of your findings.
Response: We deeply appreciate your recommendations and completely agree. We have revised the cited date to the current month and referenced the paper you recommended (Page 2, Line 71).
“The literature search was conducted using the PubMed database and included articles published up to December 2020”
We have tried our best to make the necessary changes; however, please let you know if something is still unsatisfactory.
Comment 2
Riboflavin: Among the investigated CIMs, riboflavin accumulated the greatest research interest in pediatric migraine. Your findings were indeed indicative of a potential beneficial effect of riboflavin, but additional RCTs are required. An uncontrolled trial investigating the use of riboflavin in children and adolescents was recently published and should be included (Das R, Qubty W. Retrospective Observational Study on Riboflavin Prophylaxis in Child and Adolescent Migraine. Pediatr Neurol. 2020 Sep 24;114:5-8. doi: 10.1016/j.pediatrneurol.2020.09.009. Epub ahead of print. PMID: 33189027.).
Response: We would like to thank you for this valuable paper. We have included it in the table and added a relevant discussion in the text (Page 7, Lines 161–169).
A recent retrospective study showed the efficacy of 100 and 200 mg of riboflavin in 42 patients (aged 6–18 years old): 38 with migraines, 2 with new daily persistent headaches, and 2 with chronic post-traumatic headaches [29]. There was a significant decrease in the number of days with headaches (frequency) after 2 to 4 months (11.07±10.52 days) compared to baseline (21.90±9.85 days). The mean headache intensity decreased from 8.85±6.41 to 2.30±2.51 (on a scale of 0 to 10; P < 0.001), and the headache duration significantly decreased from 18.23±17.07 to 10.18±10.49 hours; P < 0.001). A positive correlation was observed between the efficacy of riboflavin and reduced use of acute medications (rs ¼ 0.304; P ¼=0.05) [29].
Comment 3
Coenzyme Q10: Summary: The authors of the included RCT conclude ‘that children and adolescents with migraine improved over time with multidisciplinary, standardized treatment regardless of supplementation with CoQ10 or placebo’. However you conclude that ‘evidence appears to support the efficacy of coenzyme Q10 in preventing migraine in children and adolescents’. Observational studies are of limited value in case an RCT has been published, therefore, you should base your conclusions on the results of the RCT. According to the results of the included RCT, Q10 is similarly efficacious to placebo, therefore, there is no evidence suggesting its preventive effect. Please reformulate your conclusion.
Response: Thank you for your important feedback. We have revised the following sentences in the text:
(before correction) Preliminary evidence appears to support the efficacy of coenzyme Q10 in preventing migraine in children and adolescents; however, the evidence remains insufficient for a definitive conclusion to be drawn. No serious adverse events of coenzyme Q10 have been reported so far, and therefore, it may be used as a migraine prophylactic in children.
to
Despite coenzyme Q10 treatment not being associated with any serious adverse events, there is insufficient evidence to draw definitive conclusions on its efficacy (Page 9, Lines 230–231).
Comment 4
Magnesium: Summary: Wang et al. (2003) conclude that ‘This study does not unequivocally determine whether oral magnesium oxide is or is not superior to placebo in preventing frequent migrainous headache in children’. These conclusion was reached because the authors ‘were not able, however, to show that the slopes of the 2 lines were significantly different from each other’. On the other hand, Gallelli et al. (2013) performed a single blind study, in which physicians rather than participants were blinded: ‘Physicians who were blinded to the treatment assessed the overall clinical response during the study’. In view of the awareness of the participants, it is probable that the efficacy of magnesium is due to a placebo effect. Therefore, according to the findings of the first RCT magnesium is equally efficacious with placebo (the second RCT is of lesser quality in view of its open-label design participant-wise). Please reformulate your conclusion that ‘magnesium is effective in preventing migraine in children and adolescents’. There is no such evidence.
Response: Thank you for your important feedback. We have revised the following sentences in the text:
(before correction) The results of the aforementioned RCT appear to indicate that magnesium is effective in preventing migraine in children and adolescents [38]. Further, magnesium prophylaxis seems to increase the efficacy of ibuprofen and acetaminophen [39]. To support these results, further research should be carried out to assess differential efficacy based on baseline magnesium levels; this would allow patients with magnesium deficiency to benefit further from increased magnesium supplementation and dietary intake.
to
The aforementioned RCT did not definitively ascertain the superiority of oral magnesium oxide over the placebo in the prevention of frequent migraine headaches in children[38]. Further research should be performed to assess the differential efficacy in comparison with the baseline magnesium levels. This would allow patients with magnesium deficiency to benefit further from magnesium supplementation and increased dietary intake. (Page 9, Lines 266–271)
Comment 5
Melatonin: Summary: You mentioned that most retrieved studies for melatonin ‘have demonstrated favorable outcomes’. Again, melatonin seems to be as efficacious as placebo according to the presented findings. Please reformulate. Liampas et al. have demonstrated that melatonin is inferior to amitriptyline (meta-analysis) in the prevention of migraine (Liampas I, Siokas V, Brotis A, Vikelis M, Dardiotis E. Endogenous Melatonin Levels and Therapeutic Use of Exogenous Melatonin in Migraine: Systematic Review and Meta-Analysis. Headache. 2020 Jul;60(7):1273-1299. doi: 10.1111/head.13828. Epub 2020 Apr 30. PMID: 32352572.). This study should be included and the inferiority of melatonin to amitriptyline should be pointed out. An intriguing recent RCT found that melatonin could be even effective in acute attacks (Gelfand AA, Ross AC, Irwin SL, Greene KA, Qubty WF, Allen IE. Melatonin for Acute Treatment of Migraine in Children and Adolescents: A Pilot Randomized Trial. Headache. 2020 Sep;60(8):1712-1721. doi: 10.1111/head.13934. PMID: 32965037.). I recommend that this finding is at least mentioned by the authors because it may be potentially useful in the understanding of the pathogenetic mechanisms involved in pediatric migraine
Response: Thank you for your comment. The pediatric articles (50;Fallah, R et al. Iranian Journal of Child Neurology. 2018, 53;Shahnawaz, K et al. Medical forum monthly. 2019,) included in the Systematic Review that you recommended have already been cited in the current article (one of which you recommended previously). We have corrected and revised the text ad follows:
(before correction) There is a limited—but growing—number of available studies on the effectiveness of melatonin for migraine prophylaxis, most of which have demonstrated favorable outcomes.
to
There is a limited—but growing—number of available studies on the effectiveness of melatonin for migraine prophylaxis, most of which are seemingly valid. Melatonin has been demonstrated to be inferior to amitriptyline in the prevention of migraine in the aforementioned studies [50,53] A recent systematic review and meta-analysis also showed that the efficacy of melatonin for migraine has not been established in adults [54]. While a recently conducted RCT reported that melatonin was effective for acute migraine attacks [55]. (Page 11, Lines 346–352)
Comment 6
PUFAs: Paragraph 3: According to the presented findings, existing evidence is suggestive that PUFAs are as effective as placebo, therefore, there is no evidence that ‘PUFAs may potentially reduce the frequency and intensity of headache attacks’. Please reformulate to reflect these findings accurately (that PUFAs have not yet proved useful in migraine prophylaxis).
Response: We have revised the text following your comment:
(before correction) Although PUFAs may potentially reduce the frequency and intensity of headache attacks, a definitive conclusion cannot be drawn due to the paucity of existing evidence.
to
However, there is insufficient evidence that PUFAs can potentially reduce the frequency and intensity of migraine attacks; therefore, a definitive conclusion cannot be drawn. (Page 12, Lines 380–382)
Comment 7
Vitamin D: Fallah et al. recently published an article comparing the use of topiramate vs. topiramate + vitamin D3 in the prophylaxis of pediatric migraine (Fallah R, Sarraf Yazd S, Sohrevardi SM. Efficacy of Topiramate Alone and Topiramate Plus Vitamin D3 in the Prophylaxis of Pediatric Migraine: A Randomized Clinical Trial. Iran J Child Neurol. 2020 Fall;14(4):77-86. PMID: 33193786; PMCID: PMC7660031.). This study should be included and the fact that CIMs with attractive safety profiles could be used in combination regimens should be stressed. Moreover, Liampas et al. recently published a meta-analysis for the serum levels of 25(OH)D among patients with migraine and found lower 25(OH)D concentrations among adults and preliminary evidence suggesting that lower concentrations may be found in children too (one study that revealed lower concentrations and one study that found equal levels of 25(OH)D compared to healthy controls were included) (Liampas I, Siokas V, Brotis A, Dardiotis E. Vitamin D serum levels in patients with migraine: A meta-analysis. Rev Neurol (Paris). 2020 Sep;176(7-8):560-570. doi: 10.1016/j.neurol.2019.12.008. Epub 2020 Mar 30. PMID: 32241571.). This study should be included to support the evidence for the potential effectiveness of vitamin D.
Response: We have included the reference in the table and added a relevant discussion in the text (Page 12, Lines 406–413).
In a recent single-blinded, randomized, clinical trial of children with migraine headaches (5–15-year-old), the combination of topiramate and vitamin D3 showed a significant effect on the reduction of monthly headache frequency (6.12±1.26 vs. 9.87±2.44, P=0.01) and disability score (19.24±6.32 vs. 22.11±7.91, P=0.02) compared with topiramate alone [70]. Additionally, in a current meta-analysis of serum levels of 25-hydroxyvitamin D3 in migraine patients, preliminary evidence has identified low levels in both adults and children[71]. The potential efficacy of vitamin D suggests that CIM with an attractive safety profile could be used in combination regimens.
Comment 8
Conclusions: Paragraph 1: ‘there is an urgent need for additional studies’, Please tone down your wording. Urgent is a very strong word.
Response: We understand your concern and have omitted the word “urgent” accordingly. (Page 13, Line 430)
Reviewer 2 Report
The authors completed all article corrections in a timely manner.
Author Response
To the Reviewer 2
Thank you for taking the time and effort to provide such insightful feedback.
I look forward to working with you again in the future.